# Preventing suicide by restricting access to Highly Hazardous Pesticides (HHPs): A systematic review of international evidence since 2017

Bruna Rubbo[1], Chao-Ying Tu[2,3,4], Lucy Barrass[1], Shu-Sen Chang[5,6], Flemming Konradsen[7], David Gunnell[1], Michael Eddleston[8], Chris Metcalfe[1], Duleeka Knipe[1,9]*

1 Population Health Sciences, Bristol Medical School, University of Bristol, Bristol, United Kingdom, 2 Department of Psychiatry, National Taiwan University Hospital Yunlin Branch, Yunlin, Taiwan, 3 Department of Psychiatry, College of Medicine, National Taiwan University, Taipei, Taiwan, 4 Institute of Health Policy and Management, College of Public Health, National Taiwan University, Taipei, Taiwan, 5 Institute of Health Behaviors and Community Sciences, College of Public Health, National Taiwan University, Taipei, Taiwan, 6 Psychiatric Research Center, Wan Fang Hospital, Taipei Medical University, Taipei, Taiwan, 7 Global Health Section, Department of Public Health, University of Copenhagen, Denmark, 8 Centre for Pesticide Suicide Prevention and Centre for Cardiovascular Science, University of Edinburgh, Edinburgh, United Kingdom, 9 South Asian Clinical Toxicology Research Collaboration, Faculty of Medicine, University of Peradeniya, Peradeniya, Sri Lanka

* dee.knipe@bristol.ac.uk

## Abstract

Suicide is a leading cause of death worldwide. A previous systematic review showed that regulations limiting access to highly hazardous pesticides (HHPs) were successful in preventing suicides. As the WHO strongly supports regulation of pesticides, we aimed to update and strengthen the evidence on the effectiveness of HHP bans. We conducted a systematic review by searching MEDLINE, Embase, and PsycINFO databases in March 2024 for manuscripts published since 2017 that investigated the effects of regulatory changes limiting access to HHPs on pesticide suicide, at the population level. Two reviewers independently screened titles and abstracts, and extracted data using a standardized form, defined a priori. The study protocol was registered in PROSPERO (CRD42023441247). All nine studies in six Asian countries showed reductions in pesticide suicide rates following HHP bans (range 28.0% to 91.9%), of which six applied time series analyses to account for trends prior to the intervention (reductions in pesticide suicide rates ranged 28.0% to 60.5%). Only five studies assessed overall suicides; of those, four reported decreases in overall suicide rates following the intervention, of which three used time series analysis (range 7.0% to 45.1%). Only one study had a low risk of bias in all domains, with five studies having high risk of bias in at least one of the domains. Restricting access to HHPs leads to declines in both pesticide and overall suicide rates. Findings from this and the previous systematic review provide strong evidence to governments and public health officials that are considering implementing bans on HHPs in order to reduce suicides. However, this review only covered studies published since 2017 and there is a need for data from other regions to investigate the generalisability of this approach.

**Data availability statement:** All data used in this systematic review were either published by the authors in the original article or available from the authors of the original articles, upon request. This manuscript contains study-level data, extracted from the published papers and we present these data in the form of figures and tables. Where available, we requested raw data from authors of these manuscripts in order to plot the figures. Our study did not generate any new data as it is a systematic review of the existing literature; we present a summary of data generated by others.

**Funding:** This work was supported by a donation from Open Philanthropy to ME. LB is funded by grant MR/W006308/1 for the GW4 BIOMED MRC DTP, awarded to the Universities of Bath, Bristol, Cardiff, and Exeter from the Medical Research Council (MRC)/UKRI.

**Competing interests:** SSC, DG, ME, CM, and DK are co-authors in one or more papers included in this study. DG, ME and FK drafted and provided technical assistance for the development and publication of a WHO resource tool for "Suicide prevention: a resource guide for pesticide registrars and regulators" (WHO, May–June, 2019) and received a small grant from the WHO for this work, paid to the University of Bristol. The funders were involved in the design, decision to publish and final version of the document; however, no original research was involved as this work was a summary of existing evidence. DG was a member of the scientific advisory group for a Syngenta-funded study to assess the toxicity of a new paraquat formulation (2002–2006); a member of the scientific advisory group for a pesticide storage project funded by Syngenta (2005–2007); and chaired the data monitoring and ethics committee (DMEC) for a Syngenta-funded trial of the medical management of paraquat poisoning (2007–2010). DG received travel costs to attend research and DMEC meetings of the Syngenta funding studies, but no other fees. DG was Former Samaritans trustee (2015–2018) and a member of Samaritans Policy and Research Committee (2015–2021), Movember's Global Advisory Committee (2019–2022), Department of Health (England) National Suicide Prevention Strategy Advisory Group, and IASP (all unpaid roles). ME received travel fees from Syngenta for collaborative studies between 2004 and 2006. ME was member of the WHO/UN Food and Agriculture Organization joint meeting on pesticide management (as a WHO expert) from 2015 to 2023.

## Introduction

Approximately 700,000 suicide deaths occur every year [1]. Suicide rates vary greatly between regions and countries, with 77% of deaths occurring in low- and middle-income countries (LMIC) [2]. Since 2013, World Health Organization (WHO) member states have been committed to actively working towards suicide prevention through the development and implementation of prevention strategies and interventions, with a target of reducing suicide rates by a third by 2030 [2].

Contrary to the pattern observed in high-income countries (HIC), where there is a strong association between suicide and long-term mental health disorders, a greater proportion of cases of suicide in LMIC result from impulsive acts, with no to low suicidal intent [3]. This is probably due to the wide availability of lethal means, which convert acts of low intent into lethal acts. In many LMIC, the most common method of suicide is by ingestion of highly toxic pesticides [4].

Restricting access to highly lethal means of suicide is an effective suicide prevention strategy [5]. This intervention works because removing access to lethal means during a time of psychological distress allows the suicidal impulse or crisis to pass and/or the individual to survive an attempt by alternative less lethal methods of self-harm, therefore increasing survival rates. Less than 5% of people who self-harm will go on to die by suicide, and this estimate is even lower in LMIC [6,7].

Measures which restrict access to highly hazardous pesticides (HHPs) are successful cost-effective ways to prevent suicide, particularly in countries where suicide rates due to ingestion of pesticides are high, such as in many LMIC [8,9]. A previous systematic review that included studies published by 2016 found that bans on HHPs can prevent thousands of deaths worldwide [10]. Following on from this review, the WHO published their recommendation that, in countries where pesticide suicide deaths occur, HHPs should be banned [5]. As countries consider and implement national and regional-level HHP bans, a synthesis of the most recent evidence published since 2017, with a particular focus on studies that used appropriate methodology, is warranted to inform governments on the effectiveness of these interventions in reducing pesticide and overall suicide deaths.

In this systematic review, we aimed to assess if changes in regulation that limit access to HHPs led to fewer pesticide and overall suicide deaths. We build on previous literature by focusing on studies that used appropriate time series analyses published from 2017 onward, to take into account secular trends prior to the interventions, since the majority of studies included in the previous systematic review on the same topic did not use appropriate methodology, limiting the strength of their conclusions [10].

## Methods

The protocol for this study was registered with PROSPERO (CRD42023441247). We report findings according to the Preferred Reporting Items for Systematic Reviews and Meta-analyses (PRISMA) reporting guidelines [11] (S1 Checklist).

### Search strategy

We searched MEDLINE, Embase, and PsycINFO databases for English-language abstracts published between Jan 1, 2017 and Apr 21, 2023. Our search start date was aligned with the end of the search period by the previous systematic review (i.e. Dec 31, 2016). We conducted the original search on April 15, 2023 and updated the search on March 6, 2024 (S1 File). We reviewed manuscripts included in published systematic reviews and manually searched the reference list of eligible studies to identify relevant manuscripts. Additionally, we requested experts examine their personal collections to identify additional articles.

## Eligibility criteria

Abstracts documenting the effects of pesticide regulatory changes on suicide by pesticide poisoning were eligible for inclusion. Interventions included any changes in regulation on pesticides, such as changes in the concentration of products or formulations, reintroduction of pesticides that were previously banned, and complete bans on sales, imports, and use of one or more products.

We included interventions conducted at the population-level (national or regional) and excluded interventions at community levels, such as the use of lockable boxes or limiting the ease with which pesticides can be purchased. We excluded initiatives by pesticides manufacturers that were aimed at increasing the safety of their products, e.g. inclusion of emetics in the formulation and training initiatives aimed at pesticide vendors [12]. Studies conducted in selected population groups were excluded, e.g. patients admitted to hospital, data from poison centres, and studies investigating specific age groups or sex.

Manuscripts published in any language were eligible for inclusion if an abstract in English was available. We included ecological, controlled randomised and non-randomised intervention, and natural experimental designs but excluded reviews, except as potential sources of relevant manuscripts.

## Screening strategy

BR and LB independently screened titles and abstracts against the eligibility criteria and discussed any disagreements until a consensus was reached (S1 Table). A third independent reviewer was consulted (DK) where there was no agreement. BR and CY independently reviewed full texts and extracted data using a data extraction form, defined a priori. Where the full manuscript was only available in Chinese, CT alone extracted data and conducted the risk of bias assessment.

## Risk of bias assessment

Consistent with the previous review, BR and CY independently assessed the risk of bias using a modified version of the risk of bias criteria for interrupted time series studies and classified studies into three categories - high, low or unclear risk of bias (see S1 File) [13]. Studies that ignored secular trends in the pre-intervention period were recorded applying an additional item in the tool.

## Data analysis

We conducted a narrative synthesis of the data due to the high level of heterogeneity between studies. Studies varied in terms of interventions, settings, type of regulatory change, type and number of pesticides restricted, and differences in the burden of lethal pesticide poisoning to the study population. Since the proportion of suicides by pesticide ingestion is higher in LMIC, we stratified our results by LMIC and HIC. We marked as 'not assessed' where data were missing.

In a post-hoc analysis, we calculated the percentage change in pesticide suicide and overall suicide rates between the pre-intervention year and the last post-intervention year available by obtaining raw data via corresponding authors of time series studies [14–19]. We supplemented these with eligible studies from the previous review [20,21]. We used these data to fit segmented linear regression models using ordinary least squares to allow a graphical representation of pre- and post-intervention suicide trends for studies with at least two years of data before and after the intervention. We analysed, summarized, and plotted data in R version 4.2.1, using the tidyverse and robvis packages [22].

## Results

We identified 3214 abstracts for screening, of which nine manuscripts met our inclusion criteria (Fig 1). One study had a low risk of bias in all domains [17], with five studies having high risk of bias in at least one of the domains [15,23,24]. Results for each domain of the risk of bias assessment are available in Fig 2. Six studies (66.7%) used time series analysis to account for secular (pre-intervention) trends in their analysis (S2 Table).

All nine studies were from Asia, of which four were from LMIC (two from India [14,15], one from China [19], and one from Mongolia [24]) and five from HIC (two from South Korea [18,23], two from Taiwan [16,17], and one from Japan [25]) (Table 1). Most studies examined changes in regulation implemented at the national level; however, one assessed bans at both national and regional-level [15] and one focused on a particular region (Inner Mongolia) [24] (S2 Table).

Only one of the studies used interrupted time series analyses [19], reporting rate ratios (RR) and 95% confidence intervals (CI) for changes in level and slope after each intervention. The authors also adjusted for autocorrelation and seasonality by including Fourier terms and various lag periods in a sensitivity analysis. Three studies [15–17] used negative binomial

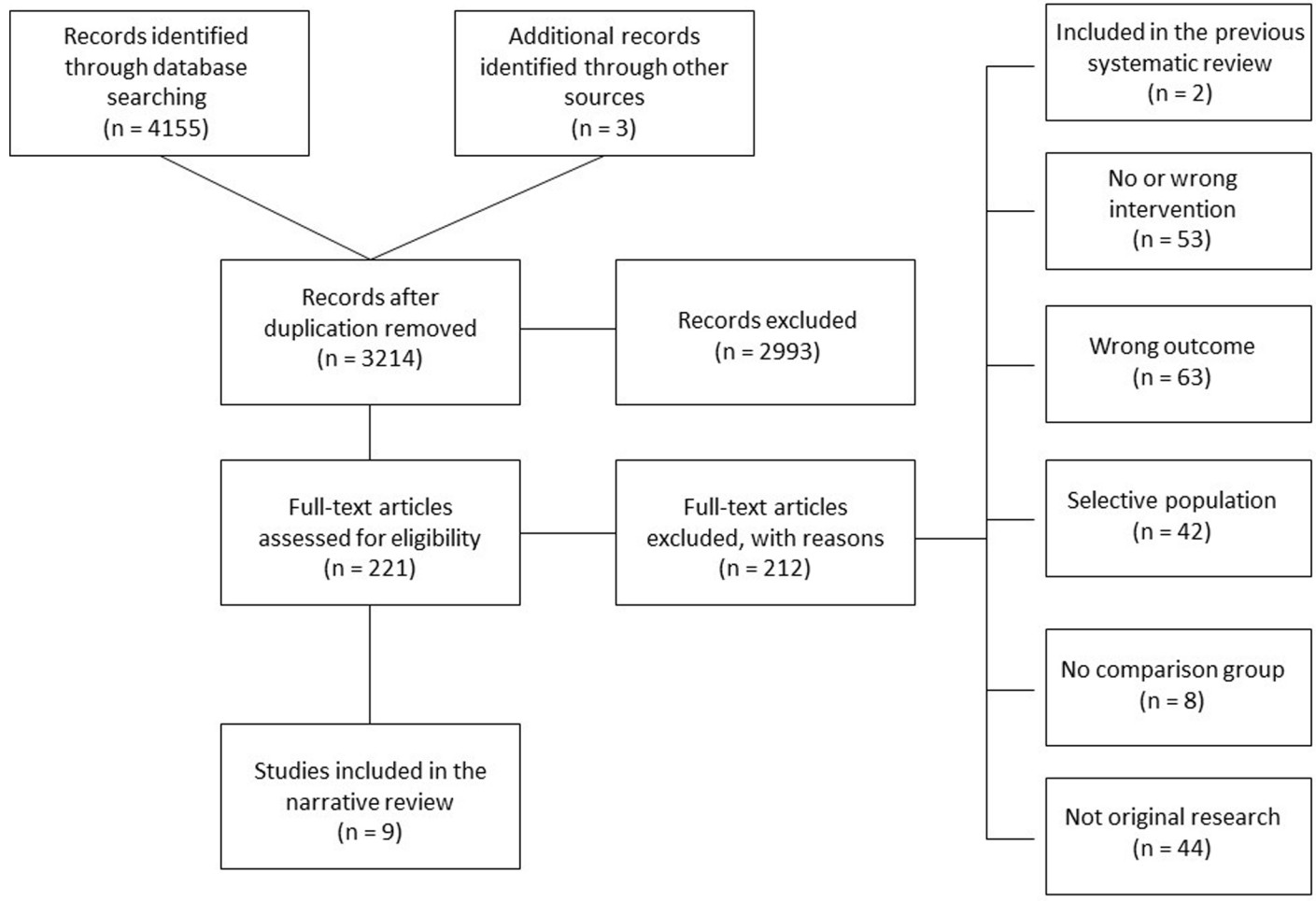

**Fig 1. Flow chart for study selection.**

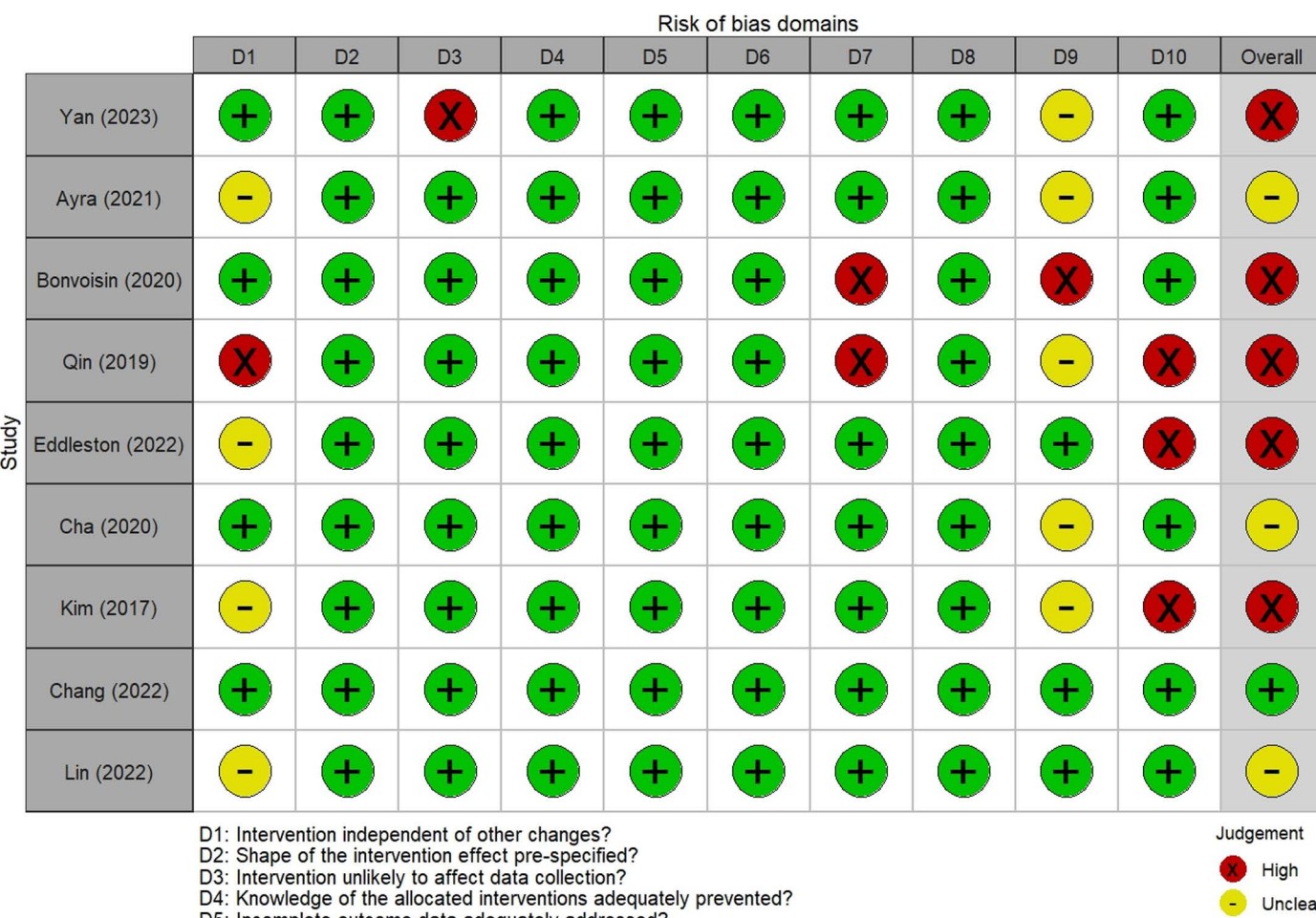

D1: Intervention independent of other changes?
D2: Shape of the intervention effect pre-specified?
D3: Intervention unlikely to affect data collection?
D4: Knowledge of the allocated interventions adequately prevented?
D5: Incomplete outcome data adequately addressed?
D6: Study free from selective outcome reporting?
D7: Study free from other risks of bias?
D8: Same source of data used for pre- and post-ban rates of pesticide suicide and legal definition of suicide?
D9: Information given about whether the regulations were effective in reducing access to key high-toxicity pesticides?
D10: Secular trends accounted for in analysis?

Judgement
X  High
-  Unclear
+  Low

**Fig 2. Risk of bias assessment for studies included in the systematic review using a modified version of the risk of bias criteria for interrupted time series studies, suggested by the Cochrane Effective Practice and Organisation of Care.**

regression to model the observed suicide rates compared to the expected rates if a ban had not occurred. One study [14] used a piecewise linear regression to break the linear trajectory into pre- and post-intervention periods, so that each trend could be calculated separately and then compared, but did not account for potential auto-correlation. Another study [18] performed a linear regression using Prais-Winsten estimation to account for auto-correlation of the first order. This method represents an extension of the ordinary least squares model, with no required assumption of independence of observations.

## Low- and middle-income countries

All four studies showed a decrease in pesticide suicide rates following changes in regulation limiting access to HHP, even when taking into account secular trends (Fig 3, Table 1) [14,15,19,24]. Changes in trends for pesticide suicide rates were observed shortly after the

**Table 1. Effect of ban on sale or import of specific pesticides for all studies eligible to be included in the review.**

| Country (main author, year) | Type of regulation change | Effect on pesticide suicides (% change between intervention and end of study period) | Effect on overall suicides (% change between intervention and end of study period) |
|---|---|---|---|
| Low - and middle-income countries | | | |
| China (Yan, 2023) [19] | Ban of 5 pesticides in December 2008<br>Ban of new paraquat parent drug (with added emetic, odorizer and colorant to paraquat) and aqueous formulations in April 2012<br>Ban of paraquat aqueous solution in July 2016 | **Decreased** – by 60.5% in 2018 | **Decreased** – by 45.1% in 2018 |
| India (Arya, 2021) [14] | National ban on endosulfan in May 2011 | **Decreased** – by 33.8% in males[*] by 39.0% in females[*] in 2014 | **Not reported** – Stated that overall suicide rates for males remained stable during the study period (2001 to 2014). |
| India (Bonvoisin, 2020) [15] | National ban on endosulfan in May 2011 | **Decreased** – by 37.3%[*] in 2015 | **Decreased** – by 10% between 2011 and 2014 |
| India (Bonvoisin, 2020) [15] | Regional ban (Kerala) on endosulfan in October 2005; Regional ban (Kerala) on 14 pesticides (including paraquat) in January 2011 | **Decreased** – by 48.3% in 2015 | **Increase** – by 3% in 2015 |
| Mongolia (Qin, 2019) [24] | Ban of 2 pesticides in 2008<br>Ban of 10 organophosphorus insecticides in 31 October 2011<br>Ban of 2 pesticides (including paraquat) in 2012 | **Decreased** – by 49% in 2015 | **Decreased** – by 33% in 2015 |
| High-income countries | | | |
| Japan (Eddleston, 2022) [25] | Paraquat 20% ion formulations restricted in 1986, with a 4·3% paraquat ion + 4·1% diquat ion combination product registered in its place. | **Decreased** – by 91.9% in 2019 | **Not reported** |
| South Korea (Cha, 2020) [18] | Ban of paraquat in October 2012. | **Decreased** – an annual percent decrease of 28% in 2014 | **Not reported** |
| South Korea (Kim, 2017) [23] | Ban of paraquat in October 2012. | **Decreased** – by approximately 50% in 2013 (extrapolated from Fig 2 in Kim et al. [22]. | **Not reported** |
| Taiwan (Chang, 2022) [17] | Nationwide ban on import and production of Paraquat from February 2018. | **Decreased** – by 37% in 2019 | **No change** in 2019 |
| Taiwan (Lin, 2022) [16] | Nationwide ban on import and production of Paraquat from February 2018. | **Decreased** – by 44% in 2020 | **Decreased** – by 7% in 2020 |

[*]Data were not available for the year prior to the intervention; therefore values from the year of intervention were used to calculate the percentage change.

implementation of the bans (Fig 3). The three studies that reported on overall suicides found a concomitant decrease in suicide rates following the intervention. The only study that did not report on overall suicides provided raw data, which, when plotted, showed a decrease in overall suicides following the intervention (Fig 3) [14]. Data from previously identified studies also showed a decrease in both pesticide and overall suicide rates following a ban on: a) paraquat, dimethoate and fenthion in Sri Lanka between 2008 and 2011, and b) all WHO Class I [26] HHPs in Bangladesh in 2000 (Fig 3) [20,21].

There was a 60.5% decrease in pesticide suicide rates in China (Table 1) from 2006 to 2018, with a steeper decline observed following the first and the third defined intervention points, when complete bans on five key organophosphorus HHPs and the sale/use of aqueous paraquat formulations were implemented, respectively [19]. The decreasing trend in pesticide suicide rates flattened after the second defined intervention point in 2012, following a ban on a new paraquat parent drug (with added emetic, odorizer and colorant) and aqueous solution (Fig 3). Pesticide rates decreased by 57.2% in areas of low urbanisation and 67.3% in areas of high urbanisation. However, before the three defined intervention points, the pesticide suicide

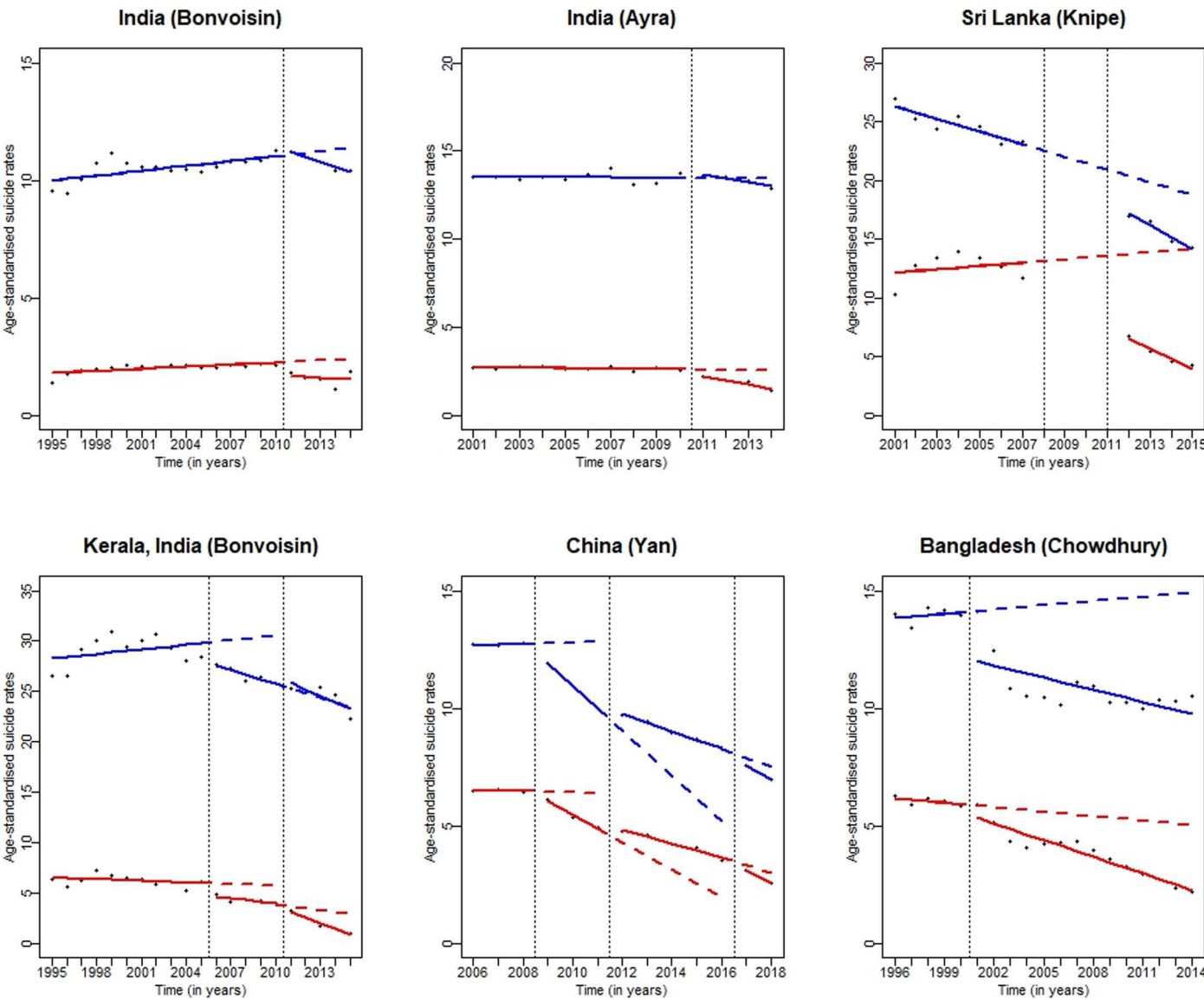

**Fig 3. Graphical representation of changes in pesticide and overall suicide deaths for studies from low- and middle-income countries.** The black dots represent the observed data points. The red solid lines represent the pre- and post-interrupted trend lines for pesticide suicide rates, and the red dashed lines represent the counterfactual trend for pesticide suicides following the intervention. The blue solid lines represent the pre- and post-interrupted trend lines for overall suicide rates, and the blue dashed lines represent the counterfactual trend for overall suicides following the intervention. The vertical dashed line shows the time of the intervention. This figure also includes data extracted from two relevant studies published in the 2017 review (ref). Please note that the scale for the Y-axis is different in each figure, as the age-standardised suicide rates vary considerably from country to country.

rate in high-urbanisation areas was already declining, and this trend did not exhibit further decreases following the pesticide bans. Conversely, in low-urbanisation areas, the pesticide suicide rate had been increasing, but the trend reversed after the bans on five organophosphorus HHPs and the sale/use of aqueous paraquat in 2008 and 2016 (S2 Table) [19].

The two studies from India covered the same regulatory change at the national level, i.e. an endosulfan ban in May 2011 [14,15]. National pesticide suicide rates decreased by 33.8%-39% in 2014, when accounting for secular trends (Table 1, Fig 3). In Kerala, a state in the southwest

of India, pesticide suicide rates were reduced by 21% between 2006 and 2010, following a regional ban on endosulfan in 2005 [15]. However, a slight increase in overall suicides was observed following the regional ban in the same period [15] (Fig 3). Unemployment rates remained relatively stable and gross domestic product per capita increased by approximately 8% per year throughout the study period; neither of which seemed to have affected suicide rates (S2 Table) [15].

The study from Inner Mongolia investigated three separate bans on two insecticides in 2008, 10 organophosphorus insecticides in 2011, and two pesticides (including paraquat) in 2012 [24]. Pesticide suicides decreased by 49%, with a concomitant decrease in overall suicide rates by 33%. However, the study did not take into account secular trends in the analyses. Additionally, the authors report several social policies aimed at improving living standards and reducing poverty levels in Inner Mongolia during the time when the third ban was implemented, which could have affected suicide rates (S2 Table) [24].

## High-income countries

All five studies showed a decrease in pesticide suicide rates following bans on HHPs (Fig 4) [16–18,23,25]. Only the study from Taiwan showed a concomitant decrease in overall suicide rates (by 7%) following the complete ban on paraquat in 2020 (Table 1 and Fig 4) [16]. Overall suicide rates (or trends) were not reported for South Korea; however, the authors provided raw data, which, when plotted, showed a reversal from an upwards trend in the pre-intervention period (Fig 4) [18,23].

The two studies from South Korea investigated the impact of cancelling the re-registration of paraquat in November 2011, which was followed by a complete ban on sales in October 2012 [18,23] (Table 1). Paraquat was responsible for a large proportion of pesticide poisoning deaths (61%) and had a case fatality of 43% in South Korea [27]. After adjusting for several socio-demographic variables, the post-ban period was associated with a 39% lower risk of pesticide suicides compared with the pre-ban period [23] (S3 Table). Cha et al. [18] showed that pesticide suicide rates decreased annually by approximately 28% in 2011–2014. The association between the paraquat ban and pesticide suicide rate was marginal in a Prais–Winsten regression analysis that adjusted for the amount of pesticides sold, the proportion of people involved in farming, unemployment rate, dependency ratio, divorce rate, the 1997 Asian economic crisis, and the 2008 Great Recession [18]. In contrast to the prominent reduction in pesticide suicide rates in rural areas after the paraquat ban, the decreasing trend in urban areas did not change after the ban [18].

The two studies from Taiwan evaluated the impact of the ban on the import and production of paraquat in February 2018 and the ban on the sale and use of paraquat in February 2020 (Table 1) [16,17]. The proportion of suicides due to pesticide poisoning was 12.1% (2011–2017), with paraquat responsible for 5% of all suicides in Taiwan before the bans [17]. Both studies showed a fewer-than-expected pesticide suicide rate following the 2018–2020 paraquat ban; 37% fewer-than-expected number of pesticide suicides in 2019, mainly due to a 58% fewer-than-expected suicides by paraquat ingestion [17], and 44% fewer-than-expected number of pesticide suicides in 2020 [16]. There was no difference in overall suicide rates up to 2019 but when data from 2020 was included in the model, they found a concomitant 7% fewer-than-expected overall suicides in 2020 [16]. The decrease in pesticide suicides was more marked among, rural residents, males, and middle-aged to older individuals compared to their counterparts [16,17].

The study from Japan assessed the impact of restrictions on sales and use of a 20% liquid formulation of paraquat, which accounted for 82.6% of deaths due to pesticide poisoning, and the subsequent introduction of a 4.3% paraquat ion with 4·1% diquat ion combination product in 1986 to replace the more toxic 20% liquid paraquat formulation [25]. Overall, there was

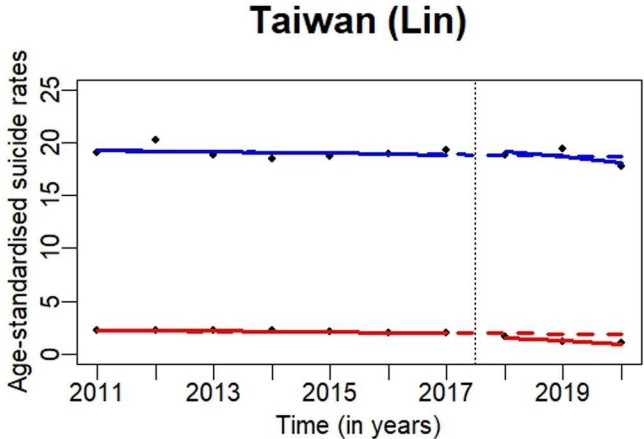

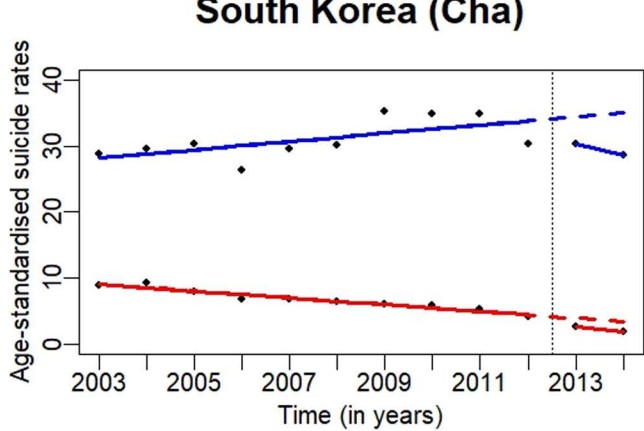

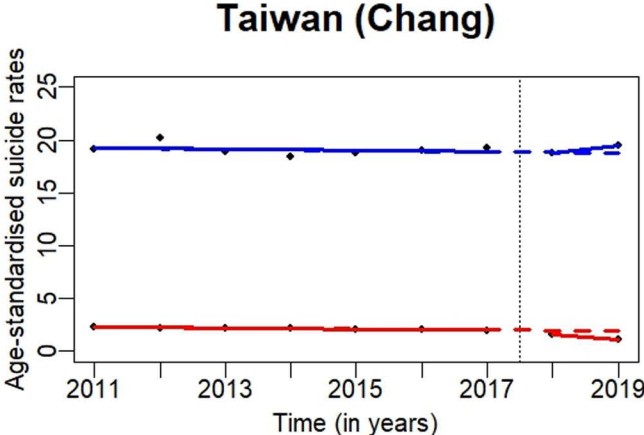

**Fig 4. Graphical representation of changes in pesticide (in red) and overall (in blue) suicide deaths for studies from high-income countries.** The black dots represent the observed data points. The red solid lines represent the pre- and post-interrupted trend lines for pesticide suicide rates, and the red dashed lines represent the counterfactual trend for pesticide suicides following the intervention. The blue solid lines represent the pre- and post-interrupted trend lines for overall suicide rates, and the blue dashed lines represent the counterfactual trend for overall suicides following the intervention. The vertical dashed line shows the time of the intervention. Please note that the scale for the Y-axis is different in each figure, as the age-standardised suicide rates vary considerably from country to country.

a reduction of pesticide suicides by 91.9%, from 8.6% of all suicides in 1985 to 0.7% in 2019 (Table 1). However, the study did not account for secular trends or assess changes in other factors concomitant to the intervention that might have affected suicide rates.

## Summary of studies reported by Gunnell et al. [10]

In the previous systematic review, Gunnell et al. [10] described 27 studies where changes in regulation for HHP had taken place in 16 countries, which included data up to 2016 [8,20,21,28–51]. Five countries were LMIC (Sri Lanka, Bangladesh, Colombia, Jordan, and India) and 11 were HIC (Greece, South Korea, Taiwan, Denmark, UK, Finland, Germany, Hungary, Ireland, Japan, and USA).

Implementation of regulatory changes limiting access of HHPs led to a reduction in pesticide suicide in most countries. In Greece, there were no changes in pesticide suicide rates

following the withdrawal of parathion and monocrotophos in Crete in 2003 [34]. Reductions in rates were observed in several European countries, such as Denmark [39], Finland [42], Germany [43], Hungary [44,45], Ireland [40,46], and England (but not Scotland) [40,41], as well as Japan [47] and the USA [49–51]. However, the picture was mixed in Ireland and in England, where there was an initial increase in pesticide suicide rates followed by a reduction in rates.

Overall suicide either decreased in three countries (i.e. Sri Lanka [8,20,31–33], Bangladesh [21], and Colombia [28]) or remained the same in one country (i.e. India [30]) following regulatory changes in all LMIC that reported on suicide rates. Results from HIC were mixed, with data from some countries showing an increase in overall suicides (i.e. Greece [34], Taiwan [37,38], UK) [40,41], Ireland [40,46], Finland [42], and USA [49–51]).

However, only five studies used appropriate time series analyses to take into account secular trends prior to the implementation of regulatory changes in access to HHPs [8,20,21,28,36]. These studies were conducted in four LMIC counties (i.e. Sri Lanka, Bangladesh, Colombia) and in one HIC (i.e. South Korea). All but the study from Colombia [28] reported decreases in pesticide suicide and overall suicide rates following bans on HHPs.

Two studies published after 2017 were included in the previous systematic review of the same topic [20,21]. These studies were unpublished at the time of the publication of the review; however, the authors had access to the findings from these and therefore decided to include them in their systematic review. Since they were published within our study period and conducted time series analyses, we used the raw data from the two studies to produce graphs representing the pre- and post-intervention periods (Fig 3).

In Sri Lanka, a phased ban on dimenthoate and fention was implemented between 2008 and 2011, and a phased ban on paraquat was implemented between 2009 and 2012. Pesticide suicide rates dropped by 50% and overall suicide rates by 21% between 2011 and 2015 [20]. Using negative binomial regression, they found that the pesticide and overall suicide rates after the intervention were lower than what would be expected should the previous trends continue [20]. The proportion of suicides due to pesticides decreased from 38% in 2000 to 30% in 2015.

In Bangladesh, organophosphorus insecticide HHPs were fully banned at the end of 2000, when all WHO class Ia and Ib pesticides were banned [21]. These included dichlorvos, dicrotophos, disulfoton, ethyl parathion, methyl parathion, mercury compounds, and monocrotophos phosphamidon. Using negative binomial regression to account for secular trends, they reported a decrease in pesticide suicides by 65.1% and in overall suicide rates by 25.0% between 1996 and 2014, lower than what would have been expected should the bans have not occurred [21]. The proportion of suicides due to pesticides decreased from 44.7% to 20.9% during the same period.

## Discussion

This review builds on and strengthens the evidence presented in a previous systematic review on the same topic by summarising findings from recent studies [10]. Gunnell et al. [10] described changes in regulation for HHP in 16 countries and this review identified two additional countries where bans on HHPs were implemented (i.e. China and Mongolia). There were four countries that were included in both reviews (India, Japan, South Korea, and Taiwan); however, only the studies conducted in Japan and South Korea assessed the same regulatory change. All studies included in this systematic review showed reductions in pesticide suicide rates following changes in regulation aimed at restricting access to HHPs.

Two thirds of the studies (i.e. six studies) in this review applied time series analyses to account for pesticide suicide trends prior to the intervention, in contrast with the previous systematic review on the same topic, where only 18% (i.e. five studies) applied time series

analyses [10]. When pre-intervention trends are not taken into account, decreases in rates following an intervention could simply be due to a continuation of existing trends, unrelated to the intervention itself. Therefore, appropriate time series models are needed in order to draw conclusions on whether an intervention has been effective. Importantly, findings from this systematic review not only built on the findings from the previous review by including recently published studies but also strengthened the quality of the evidence available, as the majority of the studies included in this systematic review used appropriate methodology to assess the effect of changes in regulations limiting access to HHPs on pesticide suicide rates.

When considering the totality of the evidence, particularly from studies that used appropriate time series methods in either one of the systematic reviews, there was a considerable reduction in both pesticide and overall suicide rates following the ban on HHPs, even when accounting for secular trends. The decrease was particularly striking in Sri Lanka (50% for pesticide and 21% for overall suicides), Bangladesh (65% for pesticide and 25% for overall suicides), China (60% for pesticide and 45% for overall suicides), India (37% for pesticide and 10% for overall suicides), and Taiwan (44% for pesticide and 7% for overall suicides) (Figs 3 and 4, S3 Table), all of which used appropriate time series analyses methods. Findings from Gunnell *et al.* [10] also showed decreases in pesticide and overall suicide rates in both HIC and LMIC; however, few of those studies used appropriate time series analyses [10], a considerable limitation of the previous review.

Even when appropriate methods were employed, other factors, such as economic growth and urbanization levels, might have had an effect on pesticide and overall suicide rates. Of the nine studies included in this systematic review, seven assessed changes in these factors [15–19,23,24], of which five used time series analysis [15–19]. Only the study from Mongolia identified events, other than the bans themselves, which could be responsible for the decline in suicide rates [24]. The two studies included in the previous review also assessed other factors that might have an effect on suicide rates. In Sri Lanka, the reduction of unemployment rates and improvement in health care services do not explain the step change observed in suicide deaths [20]. In Bangladesh, unemployment levels remained reasonably stable, including size of workforce involved in the agricultural sector, and changes in alcohol consumption did not occur at the same time as the reduction in pesticide suicides was observed [21].

It has previously been suggested that means restriction can lead to method substitutioni.e. replacement of one highly lethal method of suicide by another, such as switching from pesticide ingestion to hanging [14,32]. We find limited evidence of this, which is consistent with other means restriction interventions [52]. Means restriction is most effective in reducing overall suicides when these measures target the methods that are commonly used. Importantly, HHP bans are more cost-effective in countries where a high proportion of overall suicides are due to pesticide suicide [9]. Overall suicide rates decreased to a greater degree in LMIC included in this and the previous review following national pesticide bans, compared to HIC. This is likely because a higher proportion of suicide deaths was due to pesticide self-poisoning in included studies from LMIC (15–51%) than HIC (8–12%). Studies from China [19], South Korea [18], and Taiwan [16,17] showed that pesticide bans have a more pronounced effect on reducing pesticide and overall suicide rates in rural areas than urban areas. Additionally, pesticides bans were only effective when they targeted compounds that were used for suicide and were associated with a high case fatality. In countries where bans were implemented on pesticides products which were not responsible for suicide deaths, reductions in suicide were not observed (e.g. Taiwan from 1987 to 2010, where bans on selected pesticide products other than paraquat were implemented) [10]. This suggests that the effectiveness of pesticide bans in suicide prevention is associated with pesticide availability and with those pesticides that are responsible for the suicide deaths.

Our review has several strengths. We focused on studies that used time series analyses to assess the impact of national and/or regional-level HHP bans on pesticide and overall suicide rates. Our graphical representation of findings was designed to support translation and interpretability of findings. Our review is contemporaneous and, combined with the previous review on this topic, provides an up-to-date summary of evidence for policy makers to consider.

The review should be interpreted in light of its limitations. We restricted searches to those which had an English language abstract and to a selective few databases, and one of the manuscripts was assessed by one reviewer alone. These restrictions may have contributed to no studies being identified in Africa and South and Central America in particular. We mitigated against this by reference screening of all included studies and contacting experts in the field. The lack of studies are therefore likely to be due to a true absence rather than biased searches. As true for all reviews, we are limited by the quality of the data included and the reporting of findings from the original studies. This is a particular concern for studies using suicide data derived from official government sources in countries where suicide is considered a controversial topic due to political, religious, and social sensitivities, leading to under-reporting of deaths [53]. Our review only included studies published since 2017 and, therefore, is limited to the most recent studies on the topic. Therefore, our findings should be interpreted alongside results from the previous review by Gunnell et al. [10], which summarised the literature up to 2017. Lastly, although we performed an extensive search including personal libraries from colleagues who have good knowledge of the field, it is possible that our search has missed studies.

In conclusion, bans restricting access to HHPs can lead to significant declines in both pesticide and overall suicide rates, providing further evidence to policy makers and public health officials that are considering implementing these interventions in order to reduce suicides, in line with the WHO's LIVE LIFE programme [5] and the WHO resolution in mental health interventions [2]. This review builds on evidence of the effectiveness of HHP bans as we showed that, even when accounting for pre-existing trends, there was a marked reduction in suicide rates following the intervention, particularly in countries where pesticide suicides account for a significant proportion of all suicides.

## Supporting information

**S1 Checklist. PRISMA checklist.**
(DOCX)

**S1 File. Additional methods.**
(DOCX)

**S1 Table. Studies identified in the systematic search, with reasons for exclusion.**
(DOCX)

**S2 Table. Study main characteristics.**
(DOCX)

**S3 Table. Additional information on type of regulation change, their effect on pesticide suicide and overall suicides, and methods applied for data analyses.**
(DOCX)

## Acknowledgments

The authors would like to thank the following for providing the raw data used to produce Figs 3 and 4 of this manuscript: Vikas Arya (Western Sydney University, Australia), Shiwei Liu (Chinese Center for Disease Control and Prevention, China), and Minjae Choi (Korea

University College of Medicine, Republic of Korea). The following co-authors also provided raw data used to produce Figs 3 and 4 of this manuscript: Duleeka Knipe (University of Bristol, UK), Chao-Ying Tu and Shu-Sen Chang (National Taiwan University, Taiwan).

## Author contributions

**Conceptualization:** Bruna Rubbo, Shu-Sen Chang, Flemming Konradsen, David Gunnell, Michael Eddleston, Chris Metcalfe, Duleeka Knipe.

**Data curation:** Bruna Rubbo, Chao-Ying Tu, Lucy Barrass.

**Formal analysis:** Bruna Rubbo.

**Funding acquisition:** Michael Eddleston.

**Investigation:** Bruna Rubbo, Chao-Ying Tu, Lucy Barrass.

**Methodology:** Bruna Rubbo, David Gunnell, Michael Eddleston, Chris Metcalfe, Duleeka Knipe.

**Project administration:** Duleeka Knipe.

**Software:** Bruna Rubbo.

**Supervision:** Chris Metcalfe, Duleeka Knipe.

**Validation:** Chao-Ying Tu, Chris Metcalfe, Duleeka Knipe.

**Visualization:** Bruna Rubbo.

**Writing – original draft:** Bruna Rubbo.

**Writing – review & editing:** Bruna Rubbo, Chao-Ying Tu, Lucy Barrass, Shu-Sen Chang, Flemming Konradsen, David Gunnell, Michael Eddleston, Chris Metcalfe, Duleeka Knipe.

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
