## [Decision Letter · Decision Letter 0]

22 Aug 2024

PGPH-D-24-01535

Preventing suicide by restricting access to highly hazardous pesticides (HHPs): an updated systematic review of international evidence

Dear Dr. Rubbo,

Thank you for submitting your manuscript to PLOS Global Public Health. After careful consideration, we feel that it has merit but does not fully meet PLOS Global Public Health’s publication criteria as it currently stands. Therefore, we invite you to submit a revised version of the manuscript that addresses the points raised during the review process.

We look forward to receiving your revised manuscript.

Kind regards,

Kathleen Bachynski, PhD, MPH

Academic Editor

Additional Editor Comments (if provided):

Thank you very much for submitting this manuscript to PLOS Global Public Health. Highly hazardous pesticides represent an important global public health issue. Both reviewers recommend revisions. Reviewer #1 advises that for this to be an updated review, rather than a sequel, all 27 primary studies included in the previous 2017 analysis should be incorporated in this manuscript as well. In addition, both reviewers make several minor suggestions. Therefore, I invite you to revise and resubmit this manuscript.

Reviewers' comments:

Reviewer's Responses to Questions

**Comments to the Author**

1. Does this manuscript meet PLOS Global Public Health’s publication criteria ? Is the manuscript technically sound, and do the data support the conclusions? The manuscript must describe methodologically and ethically rigorous research with conclusions that are appropriately drawn based on the data presented.

Reviewer #1: Yes

Reviewer #2: Yes

2. Has the statistical analysis been performed appropriately and rigorously?

Reviewer #1: N/A

Reviewer #2: Yes

3. Have the authors made all data underlying the findings in their manuscript fully available (please refer to the Data Availability Statement at the start of the manuscript PDF file)?

Reviewer #1: Yes

Reviewer #2: Yes

4. Is the manuscript presented in an intelligible fashion and written in standard English?

Reviewer #1: Yes

Reviewer #2: Yes

5. Review Comments to the Author

Reviewer #1: My main concern with this review relates to the fact that this is not an “updated review” (as the authors state) since only studies from 2017 onwards were included. Findings of the 2017 review (which was conducted by several of the same authors) are not included here – hence, this is not an update but merely a (minor) sequel. Therefore, all 27 primary studies included in the 2017 review (not just references 20 and 21 which were, at that time, unpublished) should be incorporated in this manuscript as well – so that this review reflects a comprehensive picture of *all* the evidence, not just that published after 2016. For example, the current manuscript omits data from many non-Asian countries (eg, Denmark, Finland, Germany, USA, Colombia, Jordan) that were identified in the 2017 review. This major point should be addressed in the revision.

Minor comment: the ‘findings’ section of the Abstract needs more detail. Please report effect sizes (with confidence intervals) from the primary studies and other relevant statistics, as well as risk of bias assessments of the included studies.

Reviewer #2: Thank you for the opportunity to review this manuscript. It provides an important update and I am impressed that new data was obtained through correspondence. Please see some comments:

Page 8 – “ordinal least squares” Is this meant to be ordinary least squares? Otherwise please briefly explain the analysis.

Page 13 – “We mitigated against this be reference screening of all included...” Fix from be to by.

Table 1 – In columns 3 and 4, some studies have a year and some don’t. I think including the year in all rows is useful, especially if the end of the study period represents the last year of available data following the interventions.

Figures – Use the same y axis values when comparing countries in Figures 3 and 4. Otherwise use some type of annotation to highlight when the values are different – eg when comparing China and Kerala in Figure 3, the relative slope decrease appears more dramatic in China but the absolute decrease in numbers would be smaller because of the way the axis is stretched out.

Removal midline decimals if not standard journal style.

6. PLOS authors have the option to publish the peer review history of their article (what does this mean? ). If published, this will include your full peer review and any attached files.

**Do you want your identity to be public for this peer review?** For information about this choice, including consent withdrawal, please see our Privacy Policy .

Reviewer #1: No

Reviewer #2: No

---

## [Decision Letter · Decision Letter 1]

16 Oct 2024

PGPH-D-24-01535R1

Preventing suicide by restricting access to highly hazardous pesticides (HHPs): an updated systematic review of international evidence

Dear Dr. Rubbo,

Thank you for submitting your manuscript to PLOS Global Public Health. After careful consideration, we feel that it has merit but does not fully meet PLOS Global Public Health’s publication criteria as it currently stands. Therefore, we invite you to submit a revised version of the manuscript that addresses the points raised during the review process.

We look forward to receiving your revised manuscript.

Kind regards,

Kathleen Bachynski, PhD, MPH

Academic Editor

Journal Requirements:

Additional Editor Comments (if provided):

Thank you very much for the revised manuscript. Reviewer #1 recommends minor revisions to more accurately frame this review article: "all references made in the manuscript that this would be an ‘updated’ systematic review should be revised. In the title, “an updated systematic review” should be changed to “a systematic review of international evidence since 2017”. This is an important point. Same for the abstract, where it should be explicitly mentioned that the review only includes data from 2017 onwards (both in the ‘methods’ [for search strategy] and ‘interpretation’ [as a limitation] sections of the abstract). This should also be discussed in the discussion section of the manuscript."

Reviewers' comments:

Reviewer's Responses to Questions

**Comments to the Author**

1. If the authors have adequately addressed your comments raised in a previous round of review and you feel that this manuscript is now acceptable for publication, you may indicate that here to bypass the “Comments to the Author” section, enter your conflict of interest statement in the “Confidential to Editor” section, and submit your "Accept" recommendation.

Reviewer #1: (No Response)

Reviewer #2: All comments have been addressed

2. Does this manuscript meet PLOS Global Public Health’s publication criteria ? Is the manuscript technically sound, and do the data support the conclusions? The manuscript must describe methodologically and ethically rigorous research with conclusions that are appropriately drawn based on the data presented.

Reviewer #1: (No Response)

Reviewer #2: (No Response)

3. Has the statistical analysis been performed appropriately and rigorously?

Reviewer #1: (No Response)

Reviewer #2: (No Response)

4. Have the authors made all data underlying the findings in their manuscript fully available (please refer to the Data Availability Statement at the start of the manuscript PDF file)?

Reviewer #1: (No Response)

Reviewer #2: (No Response)

5. Is the manuscript presented in an intelligible fashion and written in standard English?

Reviewer #1: (No Response)

Reviewer #2: (No Response)

6. Review Comments to the Author

Reviewer #1: My main concern with the previous version of the manuscript was the fact that this was not an ‘updated review’ but merely a (minor) sequel of the previous review conducted by several of the same authors.

I’m afraid that simply adding a short summary of the 2017 review (section ‘Summary of studies reported by Gunnell et al’) in their revision (still) does not make this an ‘updated’ systematic review.

By only including nine studies published from 2017 onwards, this manuscript still provides a very selective overview of the evidence as it omits data from many non-Asian countries (eg, Denmark, Finland, Germany, USA, Colombia, Jordan) that were identified in the 2017 Gunnell review. The short summary provided in the manuscript does not address this limitation.

Therefore, all references made in the manuscript that this would be an ‘updated’ systematic review should be revised. In the title, “an updated systematic review” should be changed to “a systematic review of international evidence since 2017”. This is an important point. Same for the abstract, where it should be explicitly mentioned that the review only includes data from 2017 onwards (both in the ‘methods’ [for search strategy] and ‘interpretation’ [as a limitation] sections of the abstract). This should also be discussed in the discussion section of the manuscript.

Reviewer #2: (No Response)

7. PLOS authors have the option to publish the peer review history of their article (what does this mean? ). If published, this will include your full peer review and any attached files.

**Do you want your identity to be public for this peer review?** For information about this choice, including consent withdrawal, please see our Privacy Policy .

Reviewer #1: No

Reviewer #2: No

---

## [Editor Report · Decision Letter 2]

4 Dec 2024

Preventing suicide by restricting access to highly hazardous pesticides (HHPs): a systematic review of international evidence since 2017

PGPH-D-24-01535R2

Dear Dr Rubbo,

We are pleased to inform you that your manuscript 'Preventing suicide by restricting access to highly hazardous pesticides (HHPs): a systematic review of international evidence since 2017' has been provisionally accepted for publication in PLOS Global Public Health.

Best regards,

Kathleen Bachynski, PhD, MPH

Academic Editor

Thank you for making the final revisions in response to the reviewer comments.